# Molecular insight into *Aspergillus oryzae* β-mannanase interacting with mannotriose revealed by molecular dynamic simulation study

Uttam Kumar Jana[1,☯,¤], Gagandeep Singh[2,3,☯], Hemant Soni[2], Brett Pletschke[4]*, Naveen Kango[1]*

**1** Department of Microbiology, Dr. Harisingh Gour Vishwavidyalaya (A Central University), Sagar, Madhya Pradesh, India, **2** Central Ayurveda Research Institute, Jhansi, Uttar Pradesh, India, **3** Indian Institute of Technology, Delhi, India, **4** Enzyme Science Programme (ESP), Department of Biochemistry and Microbiology, Rhodes University, Makhanda, South Africa

☯ These authors contributed equally to this work.
¤ Current address: Crystallography & Molecular Biology Division, Saha Institute of Nuclear Physics, Kolkata, West Bengal, India
* nkango@gmail.com (NK); b.pletschke@ru.ac.za (BP)

**Data Availability Statement:** All relevant data within the article and uploaded to Mendelay at DOI: 10.17632/k2s39zw92c.1 (https://data.mendeley.com/datasets/k2s39zw92c/1).

## Abstract

Fungal β-mannanases hydrolyze β-1, 4-glycosidic bonds of mannans and find application in the generation of mannose and prebiotic mannooligosaccharides (MOS). Previously, a MOS generating β-mannanase from *Aspergillus oryzae* MTCC 1846 (*βManAo*) was characterized and its structural and functional properties were unraveled through homology modeling and molecular dynamics in this study. The *βManAo* model was validated with 92.9% and 6.5% of the residues found to be distributed in the most favorable and allowed regions of the Ramachandran plot. Glu244 was found to play a key role in the interaction with mannotriose, indicating conserved amino acids for the catalytic reaction. A detailed metadynamic analysis of the principal components revealed the presence of an $\alpha_8$-helix in the C-terminus which was very flexible in nature and energy landscapes suggested high conformation sub-states and the complex dynamic behavior of the protein. The binding of the M3 substrate stabilized the β-mannanase and resulted in a reduction in the intermediate conformational sub-states evident from the free energy landscapes. The active site of the β-mannanase is mostly hydrophilic in nature which is accordance with our results, where the major contribution in the binding energy of the substrate with the active site is from electrostatic interactions. Define Secondary Structure of Proteins (DSSP) analysis revealed a major transition of the protein from helix to β-turn for binding with the mannotriose. The molecular dynamics of the *βManAo*–mannotriose model, and the role and interactions of catalytic residues with ligand were also described. The substrate binding pocket of *βManAo* was found to be highly dynamic and showed large, concerted movements. The outcomes of the present study can be exploited in further understanding the structural properties and functional dynamics of *βManAo*.

**Funding:** The author(s) received no specific funding for this work.

**Competing interests:** The authors have declared that no competing interests exist.

## 1. Introduction

Prebiotic oligosaccharides are non-digestible carbohydrates that confer numerous health benefits and have become a recent center of attention. These compounds are emerging as an alternative to antibiotics and may play a crucial role in mitigating the emergence of anti-microbial resistance in the host [1, 2]. Different oligosaccharides available in the market are obtained either by extraction from plants or derived by enzymatic hydrolysis of various polysaccharides. Mannooligosaccharides (MOS) are short-chains of mannose obtained after enzymatic hydrolysis of mannans using endo-β-(1, 4)-mannanase (EC 3.2.1.78). Mannan-rich substrates like guar galactomannan (GG), locust bean galactomannan (LBG) and konjac glucomannan are the major sources exploited for deriving MOS [3]. The endo-β-(1, 4)-mannanase is capable of producing different oligomers of varying degree of polymerization (DP) based on the affinity and mechanism of the enzyme [4]. Like mannans, MOS also contain (1, 4)-β-linked D-mannopyranosyl units and are comprised of mannobiose (M2), mannotriose (M3), maltotetrose (M4), mannopentose (M5), etc.

β-mannanases find diverse applications in the food and feed, biofuel, oil, coffee, paper and pulp, textile industries, etc. [5]. According to the CAZy database, the majority of β-mannanases belong to 5, 26 and 113 glycosyl hydrolase (GH) families [6]. The GH5 family is a complex family with different hydrolases, including β-mannanases which have a $(\beta/\alpha)_8$ TIM barrel structure and contain a -1 sub-site of the enzyme as sugar-binding site [7]. Although mannanases with similar catalytic activities are found in different microorganisms, their abilities to bind to oligosaccharides vary, especially in the case of fungi, as these can degrade diverse hardwoods [8]. The filamentous fungus, *Aspergillus oryzae*, enjoys a GRAS status by the USA Food and Drug Administration [9]. It is employed for various purposes, especially in the production of fermented foods and various industrial enzymes (e.g. α-amylase). *A. oryzae* produces a vast range of extracellular enzymes in both solid-state and liquid cultures, with glycosyl hydrolases being the predominant enzymes. In this context, we have reported the production and characterization of a MOS generating β-mannanase from *A. oryzae* MTCC 1846 [10]. The β-mannanase showed a high rate of conversion of mannan into MOS and the oligosaccharides were shown to confer prebiotic effects [11].

Homology modeling is a computational structure prediction tool that estimates the 3D structure of a desired protein and compares it with a similar template. It is a highly precise structural prediction method that requires less time. Thus, it is very effective for the screening of different drugs and ligands based on the model [12]. Molecular docking is used to study the interaction between a ligand and a protein at the atomic residue level. The interaction reveals the biophysical and biochemical behavior of the ligand as well as the binding site of the protein. The docking process consists of ligand conformation prediction and its position and orientation within the active site (s) of the enzyme and also helps understand binding affinity of the docked molecule [13, 14]. MD simulation has been used for the analysis of conformational rearrangements of molecules and it assists in understanding the macromolecular structure-to-function relationships between the protein and ligand complex in different environments. MD simulation represents a system with several atoms, including the protein in environments mimicking a natural one [15].

In the present study, homology modeling, molecular docking and molecular dynamics simulations of β-mannanase from *A. oryzae* were performed, and interaction with mannotriose was reported. The structural and the biophysical properties of the β-mannanase were also evaluated.

## 2. Materials and methods

### 2.1 Protein sequence retrieval and domain identification

The three-dimensional (3D) structure of β-mannanase from *A. oryzae* is not currently available in the protein data bank server. For this reason, the amino acid sequence of mannanase from *A. oryzae* (*βManAo*) was retrieved from the UniProt (http://www.uniprot.org/) database (Uniprot ID: Q2TXJ2). The prediction of different conserved domains present in *βManAo* was evaluated using Pfam (http://pfam.xfam.org/), InterProScan (https://www.ebi.ac.uk/interpro/) and the Conserved Domain Database (CDD) (https://www.ncbi.nlm.nih.gov/cdd/) database servers. The EMBL-EBI Enzyme portal (https://www.ebi.ac.uk/enzymeportal/) was used to predict the substrate binding site and active site residues of the modelled *βManAo* from *A. oryzae*.

### 2.2 System preparation, refinement and validation

A suitable template for homology modeling of *βManAo* was searched for using the pBLAST suite in the Protein Data Bank (PDB) database. Results were based on the highest alignment score percentage of identity and similarity and the lowest E-value, and the best homologous structure template was retrieved. The homology modeling of *βManAo* was executed using a Swiss model homology modeling server [16]. Energy minimization and optimization of hydrogen bonding network were carried out by 3Drefine [17]. Protein secondary structures were determined by SOPMA [18]. The percentage of identity and similarity, superimpose score, and calculation of global root mean square deviation (RMSD) between the target and template were calculated by the SuperPose server [19]. The stereo-chemical quality of the model was evaluated using a Ramachandran plot by analyzing residue-by-residue and overall structure geometry by Procheck [20]. Comparison between homology modeling, X-ray and NMR structure was predicted by ProSA and the Z-score of the model was also calculated by the webserver [21]. ERRAT was used for the statistical analysis of various atom types and non-bonded interactions of the refined model and also for plotting the error function value versus position of a 9-residue sliding window [22]. The compatibility of the atomic model with amino acid sequence by calculating a structural class of different location and environment (polar, non-polar, loop, alpha, beta, etc.) was predicted by Verify 3D [23]. The secondary and super secondary structure of the model was evaluated by Stride [24]. QMEAN was used for predicting theoretical model Z-score in comparison to a non-redundant set of PDB structures [25].

### 2.3 *In silico* physico-chemical characterization of *βManAo*

Different physicochemical characters such as molecular weight, number of positive and negatively charged amino acids, extinction coefficient, theoretical isoelectric point, aliphatic index, instability index and grand average hydropathicity (GRAVY) of *βManAo* were predicted by ExPASy's ProtParam web server tool (http://web.expasy.org/protparam/). Secondary structure, solvent accessibility, and different ontology studies such as molecular function, cellular components, and biological process were predicted by the Predict Protein server (https://predictprotein.org/).

### 2.4 Intra-atomic interaction analysis of *βManAo*

The non-covalent intra-molecular interaction including H-bonds, ionic bonds, disulfide bonds, π-cation, and π-π stacking bonds of *βManAo* at the atomic level was visualized using the Residue Interaction Network Generator (RING) web server and Arpeggio web server [26, 27]. The salt bridges in the protein structure were analyzed by the ESBRI web server [28].

## 2.5 Ligand preparation

The interaction between *βManAo* with mannotriose was investigated using the molecular docking method. The topological properties (including the surface pockets and interior cavities of the homology model) were predicted by CASTp 3.0, which exhibited the negative volumes of different binding pockets and gave secondary structure, functional sites, variant sites, and other annotations of protein residues of the enzyme [29]. 3D structure of mannotriose was retrieved in SDF format from the PubChem database (M3, PubChem ID: 3010288). The energy minimization of mannotriose was performed using the minimize structure tools, where the parameters like steepest descent steps (100), steepest descent step size (0.02 Å), conjugate gradient steps (10), conjugate gradient step size (0.02 Å) and update interval (10) were set for the minimization step. The minimizations of standard residues were performed using the AMBER ff14SB force field and other residues were executed using the Gasteiger force field.

## 2.6 Molecular docking

The molecular docking between M3 and *βManAo* was executed using AutoDock 4.2 [30]. The grid box was generated by AutoGrid and selected 3D coordinates in x, y, z-dimensions, where the active site of the enzyme as well as a large portion of adjoining surface was covered. After completing the docking process, nine probable docking structures were generated and the best combination was evaluated on the basis of minimum binding energy (Kcal/mol), number of hydrogen bonds, docking score, and other weak interactions. Results were visualized using Discovery Studio Software. The absolute binding affinity of *βManAo*-M3 complex was predicted by $K_{DEEP}$ [31].

## 2.7 Molecular dynamics simulation

In order to assess the stability of *βManAo* structure, an all-atom molecular dynamic simulation of *βManAo* and *βManAo*-M3 was run using LiGRO [32], a GUI based software that prepares the necessary files required for MD simulations using GROMACS 5.1.5 [33]. The protein was solvated in a cubic box filled with TIP3P water molecules such that the distance between two periodic images was 2 nm. The amber99sb forcefield was used for protein while the topology for the M3 substrate was generated using an integrated ACPYPE module of LiGRO with General Amber Force Field (GAFF) and bcc charges [34]. The system was neutralized by adding 150 mM NaCl. In order to remove the steric clashes, the system was energy minimized using 5000 steps of steepest descent followed by a conjugate gradient with a tolerance of 10.0 kJ/mol/nm. The system was subjected to equilibration under NVT and NPT ensembles for 1 ns each at 310.15 K and 1 bar, using a modified Berendsen thermostat and Parrinello-Rahman barostat, respectively. The verlet cutoff-scheme was used for neighbor searching with short range electrostatic and van der Waals energy cutoff values of 1.4 nm. The molecular dynamic production simulation of 100 ns under NPT ensemble was run on the high-performance computing cluster at the Indian Institute of Technology, Delhi (HPC-IITD). A PME method and LINCS algorithm were applied to correct for the long-range electrostatic interactions and to constrain the covalent bonds. A time step of 2 fs was used while the frames were updated after every 500 steps. The trajectory was visualized using VMD and Chimera and standard GROMACS tools were used to analyze the trajectory.

**2.7.1 Geometrical properties.** Different geometrical parameters such as the radius of gyration (RG), number of hydrogen bonds (NHB), root mean square deviation (RMSD), root mean square fluctuation (RMSF), and energy components such as entropy, Gibbs free energy of binding of protein and protein-ligand complex were calculated with respective programs

using GROMACS tools. The substrate binding pocket flexibility and dynamics were analyzed using TRAPP webserver [35].

**2.7.2 Principal component analysis.** Principal component analysis (PCA) is a technique to increase the interpretability of large datasets by reducing the complexity of the data and lowering the information loss from the dataset. PCA was implemented with the new uncorrelated variables using a variance/covariance matrix generated from MD trajectories. A covariance matrix was obtained from the atomic fluctuation after separating the rotational and translational movement and it was represented by the simple linear transformation in Cartesian coordinate space. Further, diagonalization of the matrix ($C_{ij}$) in PCA was obtained through an orthogonal matrix by computing on the basis of Eq (1).

$$C_{ij} = V\Lambda VT \tag{1}$$

where, $\Lambda$ = eigenvalues as diagonal entries, V = represents the related eigenvectors and T = orthogonal coordinate transformation matrix. The PCA plots for the *βManAo* and *βManAo*-M3 models were generated using the Geo-Measures [36] plugin in PyMOL employing the first two most populated principal components (PC1 and PC2).

**2.7.3 Free energy landscape analysis.** Free energy landscapes of any system explain all the conformational entities of a molecule and their roles in interacting with other molecules with respect to the spatial position, and play a key role in deciphering their respective free energy levels. The free energy landscapes (FELs) of the *βManAo* and *βManAo*-M3 models were constructed using the Geo-Measures tool employing the trajectory RG by RMSD values. Geo-Measures were employed by using the g_sham tool of the GROMACS package to generate the FEL (https://manual.gromacs.org/archive/4.6.1/online/g_sham.html). The stable energetic conformations were marked in blue, while the less stable conformations were marked in red regions.

**2.7.4 Binding free energy calculations using MMGBSA methods.** The MMGBSA module of the gmx_MMPBSA tool was used to calculate the binding energy *ΔG* from the last 100 frames from the 90-100ns interval of the trajectory [37]. The GB method used was igb = 2 with internal and external dielectric constants of 1 and 80, respectively, being applied. The entropic contributions were estimated by normal mode analysis (NMA). The estimated *ΔG* is given by Eq (3),

$$\Delta G = \Delta H - T\Delta S = \Delta G_{gas} + \Delta G_{soly} - T\Delta S \tag{2}$$

where, $\Delta G_{gas}$ = EEL + VDWAALS, represents the total gas phase energy consisting of electrostatic and Van der Waals interaction energies, $\Delta G_{solv}$ = EGB + ESURF, are the polar and non-polar solvation free energies and *TΔS* corresponds to the change in conformational entropy on binding.

**2.7.5 Define Secondary Structure of Proteins (DSSP) analysis.** The secondary structural changes in the protein with respect to the frames in the trajectory of *βManAo* and *βManAo*-M3 were measured by DSSP [38]. The secondary structure was labeled as 'H' for helix, 'E' for strands and 'C' for the coils.

# 3. Results and discussions

## 3.1 System preparation, refinement and validation

The 3D structure of *βManAo* had the maximum homology with the chain A of β-mannanase from *A. niger* with a 70.93% identity among their amino acids and belonged to the GH5 glycosyl hydrolase family (Fig 1A). The total score, query coverage, and E-value between the two

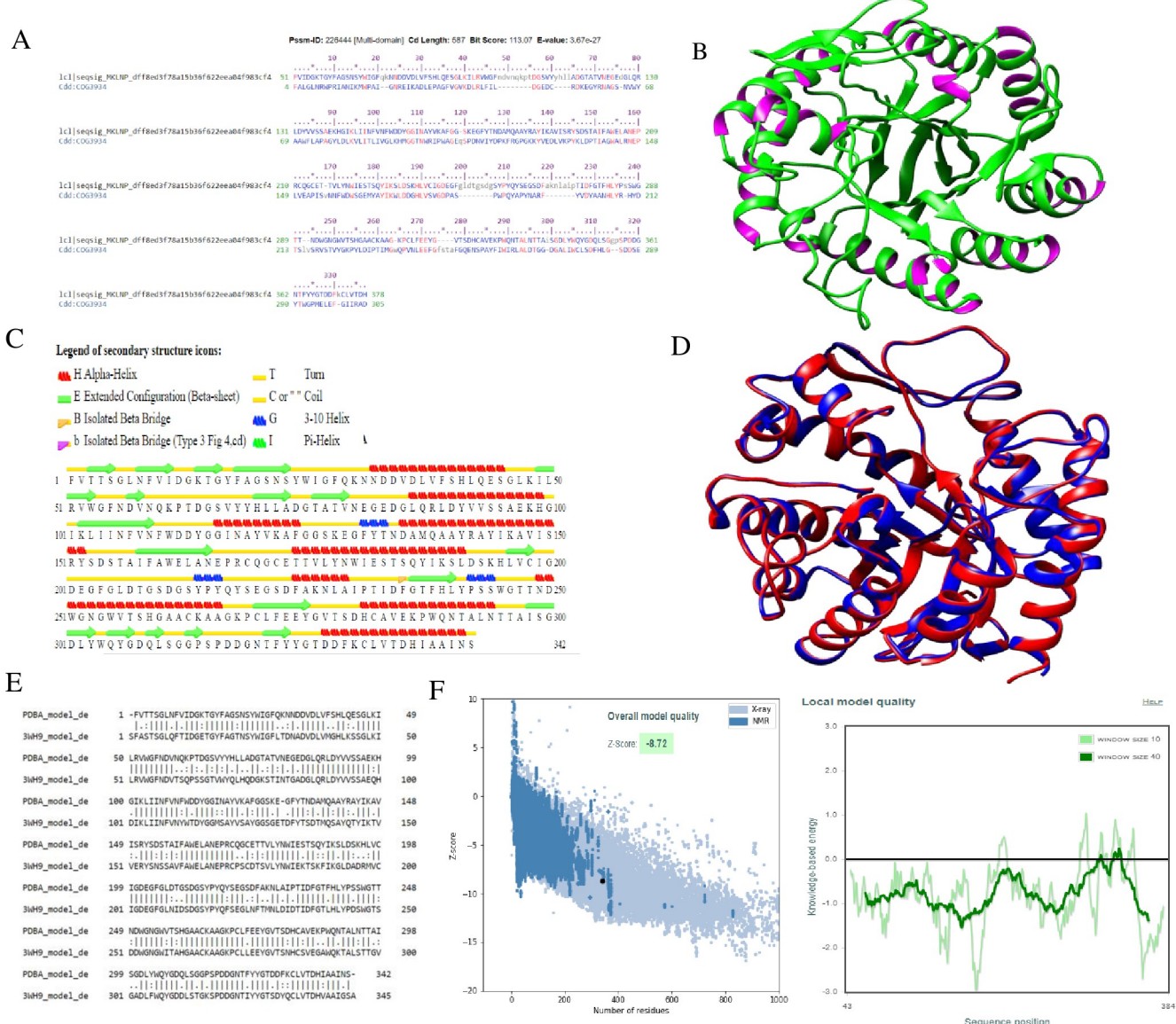

**Fig 1. Models generated for *A. oryzae* mannanase (*βManAo*) and their analysis.** (A) CDD analysis of *βManAo* denoted that the enzyme belonged to GH5 carbohydrate active enzyme family, (B) The 3Drefiner energy minimized model of *βManAo* by UCSF Chimera, (C) Predicted secondary structure of *βManAo* showed majorly α-helix followed by extended strand and beta-turn were the main structure components revealed by Stride, (D) Image of superimpose structure between *βManAo* and chain A of 3WH9, (E) Basepair wise structure alignment of *βManAo* and chain A of 3WH9, (F) Protein Structure Analysis of *βManAo*. Overall model quality and local model quality of the enzyme model showing most of the sequence in the negative energy mode.

sequences were 528, 88% and 0, respectively. These values indicated that the template of β-mannanase from *A. niger* (3WH9) was suitable for homology modeling. The predicted model of *βManAo* by model preparation and refinement by 3Drefine is shown in Fig 1B. The secondary structure of the protein as predicted by Stride (Fig 1C) showed that the percentages of the helix, extended strand, beta-turn, and the random coil were 33.94%, 18.13%, 6.22%, and 41.71%, respectively [24]. The superimposition between the two templates as analyzed by SuperPose showed that 342 atoms in the alpha carbons of both templates were superimposed and the root mean square deviation (RMSD) was 0.25 Å. The superimposed *βManAo* with

chain A of 3WH9 and pair-wise alignment are presented in Fig 1D. Similar to the NCBI protein BLAST, the identity percentage was 70%, whereas the similarity score was 83%. The base-pair wise structure alignment is shown in Fig 1E. Protein Structure Analysis (Pro-SA) is a useful tool for the refinement and validation of an experimental structure. The main parameter for Pro-SA is the Z-score which evaluates the overall structure quality, as well as provides a measure of the total energy deviation of the structure with the help of energy deviation based on random conformation. For this reason, the value of the Z-score of the protein model outside the region and the positive value of the energy plot is referred as problematic and erroneous [21]. Here, the Z-value of the X-ray structures estimated for *βManAo* was -8.72 (Fig 1F) and the maximum sequences were in the negative energy mode (Fig 1F), which corresponded to the correct model. In addition, the Z-score of the predicted model was calculated by a normalized QMEAN score of 0.5< to <1, when compared with a non-redundant set of PDB structures, which suggested the appropriateness of the model with high quality (Fig 2A).

The Ramachandran plot for *βManAo* obtained from Procheck showed that all the amino acids of the predicted model make a favorable region, where 92.9% were in the most favored regions, 6.5% in the additional allowed regions, and 0.7% in the generously allowed regions. No residues were present in the disallowed regions and R-factor was not greater than 20%, which again indicated a good quality model (Fig 2B). Furthermore, the G-factor of the model

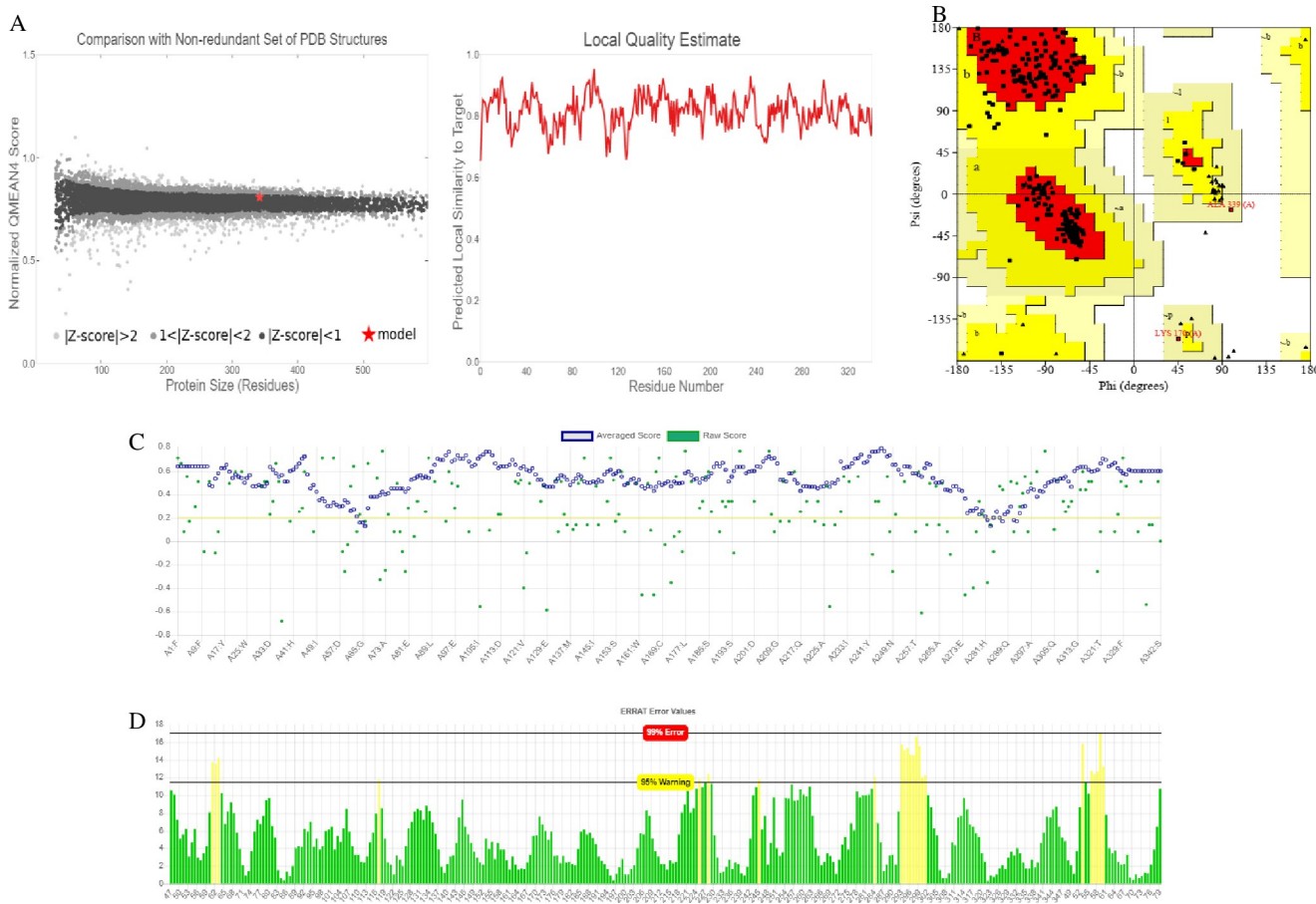

**Fig 2. Validation of the predicted *βManAo* model.** (A) QMEAN score of the predicted model and local quality estimation of the model, (B) Ramachandran plot of *βManAo* by Procheck indicated that all the amino acids residues are falling in the allowed region, (C) Assessment of 3D model of *βManAo* by Verify 3D and (D) Assessment of non-bonding interaction of different atoms by ERRAT.

was -0.58, which was low and satisfactory, as the higher negative value of the G-factor refer to the low-probability conformation with amino acid residues of the model falling in the disallowed regions of the Ramachandran plot. The 3D model assessment through Verify 3D resulted in 97.66% of the residues with an average 3D - 1D score > = 0.2. Fig 2C depicts the plot of 3D-1D score of amino acids of *βManAo*. ERRAT analysis was found to be 93.1138, which indicated a recommended statistical value for non-bonded interactions between different atom types and value of error function *vs.* position of a 9-residue a sliding window of the refined model (Fig 2D). Therefore, after the quality and refinement assessment of *βManAo*, it was concluded that the predicted model of the β-mannanase from *A. oryzae* showed stereochemical stability and can be successfully used for further analysis.

## 3.2 Physico-chemical properties of *βManAo*

The pI of *βManAo* was computed as 4.69, indicating the acidic nature of *βManAo*. The extinction coefficient of the predicted *βManAo* was 88155 $M^{-1}$ $cm^{-1}$ at 280 nm, which indicated the hydrophobic nature of the enzyme due to the presence of 54.92% non-polar amino acids. *βManAo* had two disulfide (S-S) bonds, which provided stability to the 3D structure as well as assisted in redox activity. The presence of disulfide bonds also indicated the secretory nature of the protein [39]. The instability index (II) of *βManAo* was computed to be 21.55, which showed that the protein was highly stable under the suggested *in vitro* conditions (Table 1).

Similarly, β-glucosidase from *Trichoderma* sp. had an instability index of 31.25 [40]. *βManAo* had a higher aliphatic index of 70.54, which is the reason for its activity over a wide temperature range, while a negative value of GRAVY (-0.28) indicated its more favorable aqueous interactions. As suggested by the model, *βManAo* had a high content of alanine (8.3%), glycine (10.1%) and serine (10.1%) amino acids.

## 3.3 Domain and substrate binding pocket analysis of *βManAo*

Analysis of different domains through Pfam, InterProScan and Conserved Domain Database (CDD) revealed that the protein had two conserved domains, such as Signal peptide (1–21 AA) and the main glycosyl hydrolase (GH5) ranging from 51 to 378 AA (Fig 1A). The GH5 domain displayed close homology with the chain A of 3WH9 (PDB ID), which was found to be an endo-β-1, 4- mannanase from *Aspergillus niger*. The substrate binding and active site residues of *βManAo* were predicted by EMBL-EBI Enzyme portal. The predicted substrate binding residues include Trp95, Asn207, Tyr 283 and Trp346 (on sharing similarity with β-mannanase of *Cryptopygus antarcticus*, Uniport: B4XC07) and the active site residues include Glu208 (Proton donor) and Glu316 (Nucleophile) (on sharing similarity with β-mannanase

**Table 1. Predicted physico-chemical characteristics of *βManAo* through ExPASy's protparam.**

| | |
|---|---|
| Protein ID (Uniprot) | Q2TXJ2 |
| Sequence length (No. of AA) | 386 |
| Molecular weight | 41906 Da |
| pI | 4.69 |
| Total number of negatively charged residues (ASP + GLU) | 42 |
| Total number of positively charged residues (ARG + LYS) | 24 |
| Extinction coefficient | 87780 |
| Instability index (II) | 21.55 |
| Aliphatic index | 70.54 |
| GRAVY | -0.288 |

from *Trichoderma reesei*, Uniprot: Q99036). These predictive results are in accordance with our docking results where the M3 molecule is interacting with the predicted substrate binding and active site residues.

### 3.4 Intra-atomic interactions

Among all the intramolecular actions, hydrophobic interactions and hydrogen bonds were found to be the major reason for the stability of *βManAo* (Fig 3).

A total 1072 hydrophobic interactions and 444 H-bonds were present. *βManAo* had more charged polar amino acids on the surface while non-polar side chains were buried, which made a favorable contribution to protein stability by removing the non-polar side chains from water and enhancing the London dispersion forces that result in the tight packing of protein from the interior side [41]. H-bonds are not only involved in protein-ligand interactions, but are also crucial for the conformational stability of a protein for optimum physical properties [42]. Asp, Arg, and Glu were the main amino acids involved in the ionic interactions. The endo-β-mannanase from *Arabidopsis thaliana* formed two ionic bonds with mannose at Glu252 [43]. Ionic interactions increase the thermo-stability of the enzymes, while ionic-pair networks help enzymes withstand the local environment [44]. As discussed earlier, the enzyme had two disulfide bonds at the positions Cys373-Cys324 and Cys312-Cys305 (Fig 3). The β-1,

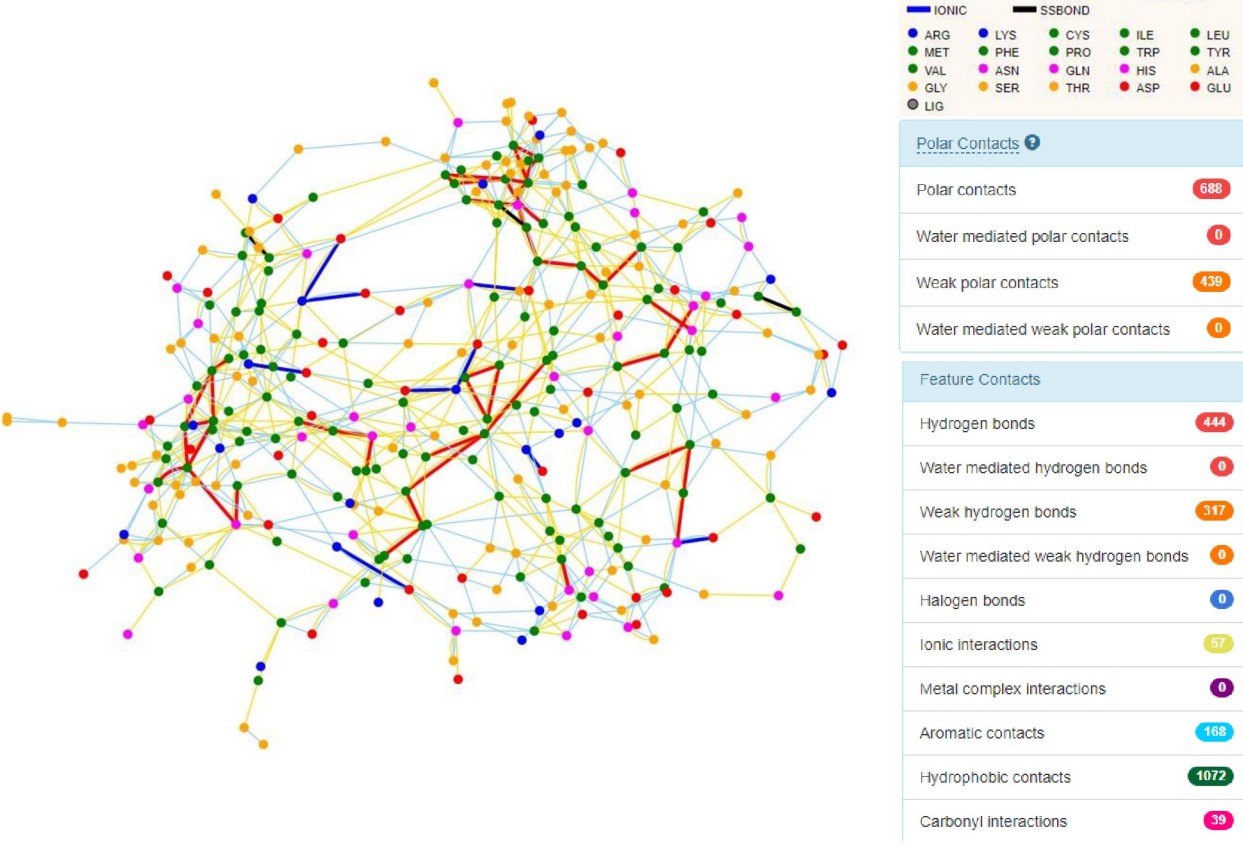

**Fig 3. Intramolecular interaction of *βManAo*.** Main interactions are designated as Van der Waals interactions (yellow), conventional H-bonds (light blue), attractive non covalent π–π stacking interactions (red), electrostatic attraction as ionic bonds (dark blue) and disulfide bridges (black). Right columns represent the total number of different interactions. *βManAo* had majorly hydrogen bonds for the stability of the enzyme with two disulfide bonds at the positions Cys373-Cys324 and Cys312-Cys305.

4-mannanases from *Trichoderma reesei* and *Podospora anserine* had four (Cys26-Cys29, Cys172-Cys175, Cys265-Cys272, and Cys284-Cys334) and three (Cys180-Cys184, Cys272-Cys279, and Cys291-Cys342) disulfide bonds, respectively [45, 46]. *βManAo* showed several weak interactions which are important for the stabilization of the enzyme as well as in the determination of its tertiary structure. Similarly, salt bridges have a pivotal role in maintaining protein stability and solubility. The amino acids, His/Asp, Lys/Asp, Lys/Glu, Arg/Asp, Arg/Glu, His/Asp and His/Glu are required to form salt bridges. Among the six salt bridges, His/Asp formed 32% of the total salt bridges, which were less than 4.0 Å. Lys/Asp residues formed 25% of the salt bridges, while Arg/Asp and Arg/Glu residues accounted for 12% of the salt bridges each, in the *βManAo* structure.

### 3.5 Molecular docking with mannotriose

The interaction between the substrate and enzyme (mannotriose, M3 and *βManAo*) was studied by molecular docking with the help of AutoDock 4.2 and visualized by BIOVIA Discovery Studio Visualizer. The list of time dependent specific amino acids of the protein interacting with the mannotriose *via* different bonds is given in S1 Table. *βManAo* interacted with M3 *via* five conventional H-bonds. The H-bonded amino acids were Glu208 (2 bonds), Asn151 and Glu244 (2 bonds) (Fig 4A and 4B).

The binding energy for M3 was -5.4 kcal/mol with a pK$_d$ value of 4.0 and a cluster RMS value of 0.00 Å. The ligand binding efficiency was found to be -0.15.

### 3.6 Molecular dynamics simulation

**3.6.1 Radius of gyration (RG).** The RG value was calculated for investigating the compactness and structural changes in the *βManAo* and *βManAo*-M3 complexes. The RG value of a protein is calculated by measuring the root mean square distance of an atom of protein in relation with the center of mass of the protein. The overall average RG value of *βManAo* was ~1.90 nm and ~1.91 nm in case of *βManAo*-M3 complex after 100 ns MD simulations were run (Fig 5A). Both the structures remained intact after 20 ns and throughout the full MD simulation run. The compactness and structural integrity of the *βManAo*-M3 complex were similar to the native protein. RMSD value of C$_\alpha$ atoms was also well correlated with the RG value. The

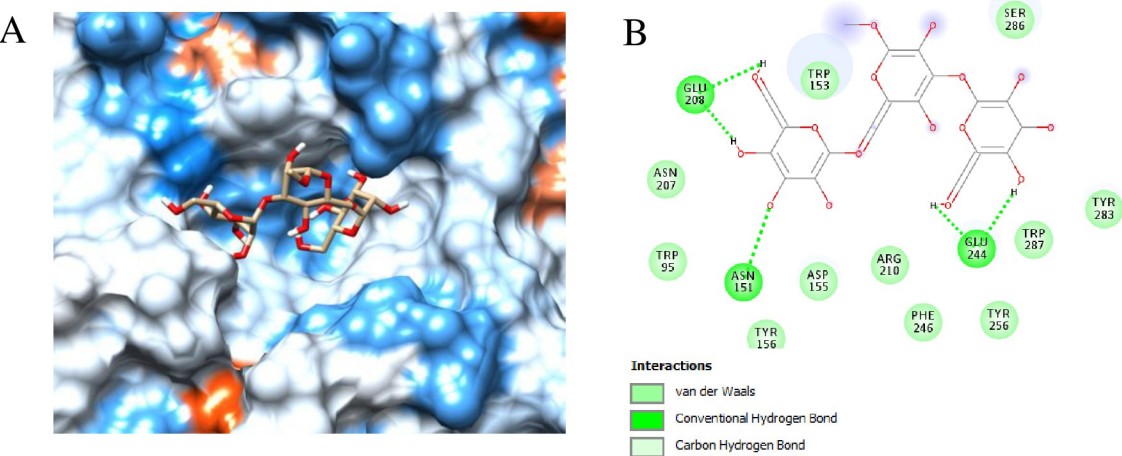

**Fig 4. Binding cavity and the 2D chemical figure of the binding site residues of *A. oryzae* β-mannanase (*βManAo*) with mannotriose (M3).** (A) Catalytic cavity of *βManAo*, (B) Active site amino acids interacting with M3 after docking.

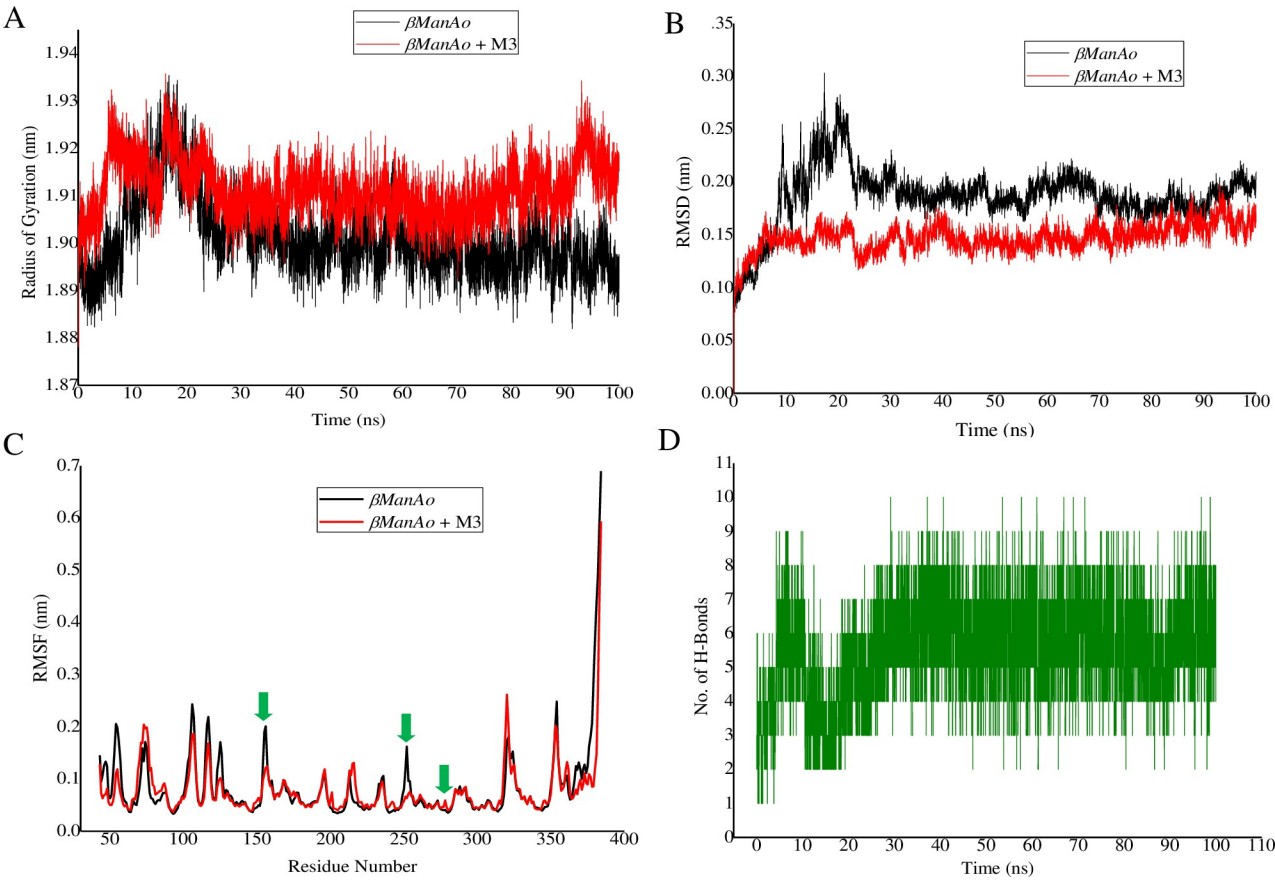

**Fig 5. Analysis of different protein dynamics parameters.** (A) Combined radius of gyration (RG) plot of *βManAo and βManAo*-M3 at 310.15 K, (B) Time dependent backbone RMSD values with respect to the starting structure of protein and protein ligand complex during MD simulations, (C) Combined RMSF plot and (D) Number of total H-bonds interacting with M3 during simulation. Glu208, Glu262 and Asn155 were the main catalytic amino acids involved in H-bonding. Color illustration: Native Protein designated as *βManAo* (black) and *βManAo*-M3 complex (red).

differences between the two structures were insignificant, which suggested that the *βManAo*-M3 complex was stable over the period and that the modeled protein folding was correct.

**3.6.2 Root Mean Square Deviation (RMSD) analysis.**   The main motive behind the MD simulation runs was to understand the stability of the protein-ligand complex after docking. The RMSD was calculated for the overall 100 ns simulation run. The RMSD value of a protein provides an insight into the structural conformation through the MD run and the complex RMSD value helps to understand the stability during the catalytic activity of the protein. The RMSD graphs of both *βManAo* and *βManAo*-M3 were generated with respect to the modeled structure during the 100 ns. The average RMSD value of *βManAo* was ~0.186 nm, whereas that of the *βManAo*-M3 complex was found to be 0.147 nm (Fig 5B). Initially, a light fluctuation in the RMSD value for *βManAo* from 10 ns to 25 ns was observed, but it was altogether absent in case of the *βManAo*-M3 docked complex. The possible reason behind the *βManAo*-M3 stability was the binding of the substrate with the active site amino acid residues located in the structure. Overall, both predicted structures, the *βManAo* and the *βManAo*-M3 complex, remained stable throughout the 100 ns MD simulation run without any significant variance between them, and most of the residues of the protein had stable conformations as minimum amino acids switched their native conformation from bends to turn. The results suggested that the catalytic residues on interaction with mannotriose displayed stability in the overall run.

**3.6.3 Root Mean Square Fluctuations (RMSF).** The RMSF of each of the 342 residues were determined and superimposed on both *βManAo* and the *βManAo*-M3 complex (Fig 5C). The average RMSF values of both showed moderate fluctuation from 0.077 to 0.083 nm. Overall, the root-mean-square fluctuation of the active site binding residues (Asp151, Glu244) of the *βManAo*-M3 complex were decreased as compared to *βManAo* (marked by a green arrow) (Fig 5C). The binding residues position between 200 to 300 amino acid sequence had low RMSF fluctuation between indicating compactness and rigidity of the catalytic cavity. The last ten C-terminal residues showed a highly fluctuating graph, possibly because of the unavailability of modeled protein structure information.

**3.6.4 H-bond interaction.** The H-bond interaction was analyzed for the docked complex of *βManAo*-M3 for 100 ns, using a standard GROMACS hbond tool as shown in Fig 5D. Glu244 formed continuous two conventional H-bonds with mannotriose as a major active site residue. The ligand-protein complex was stabilized with an average of six or seven H-bonds in the 100 ns run. In addition, Glu208, Glu262 and Asn155 also participated in H-bonding. Besides these residues, Ser286, Ser285, Arg210 and Trp287 did not form H-bonds, but were found to actively participate by Van der Waals and electrostatic interactions in stabilizing the enzyme-ligand complex.

**3.6.5 Binding pocket dynamics.** The overall dynamics of the protein active site was monitored as a function of the simulation time frame of 100 ns. The *βManAo* displayed a flexible active site that varied in volume ranging from 293 $Å^3$ to 599 $Å^3$ (Fig 6) as the M3 molecule moved from the initial (docked) position to probable active site release position (Fig 7A) over the simulation time frame of this study.

The residue Tyr283 acts as an anchor point between the initial and probable substrate release conformational state of M3 (Fig 7B). The comprehensive representation of overall appearing and disappearing pocket is shown in (Fig 7C).

**3.6.6 Principal component analysis (PCA).** PCA was used for analyzing the dynamics of protein and also to understand the dynamic properties in light of the MD simulation run. Two eigenvector projections (PC1 & PC2) for *βManAo* and *βManAo*-M3 are shown in Fig 8A & 8B.

These principal components showed maximum molecular motion in the protein as a function of distance. In case of *βManAo* (PC1), the initial motion was anisotropic. This may be attributed to the high movement with different directions of $\alpha_8$-helix present in the C-terminus of the protein [47]. These movements resolved with increasing time and were absent in the case of enzyme-ligand complex, which indicated that mannotriose binding increased the stability of the overall protein structure. The trace values of the two eigenvectors were more negative in the case of *βManAo*-M3 complex compared to *βManAo*, which led to more flexibility in the $C_\alpha$ atom. The binding of the mannotriose expanded the protein and changed the overall motion of the protein.

**3.6.7 Free energy landscape (FEL) analysis.** The free energy landscape (FELs) was generated as a function of RMSD and RG values obtained through the MD trajectory. The free energy profiles of *βManAo* and *βManAo*-M3 complex were similar in nature. In the case of *βManAo*, there were multiple local minima and global minima with a Gibbs free energy difference of 5–6 kJ/mol while there was a single global energy minimum for the *βManAo*-M3 complex. Therefore, the FEL appeared more funnel-bottom like than the *βManAo*-M3 complex (Fig 9A & 9B).

These results led to the conclusion that *βManAo* had high conformation sub-states, more complex dynamic behavior and was richer in conformational diversity but binding with mannotriose these characters were subsidized. These results are similar to the previous report by Sang et al. [48], who suggested that a psychrophilic proteinase K had more local energy minima and a funnel-like free energy landscape than a mesophilic proteinase K. The FELs are

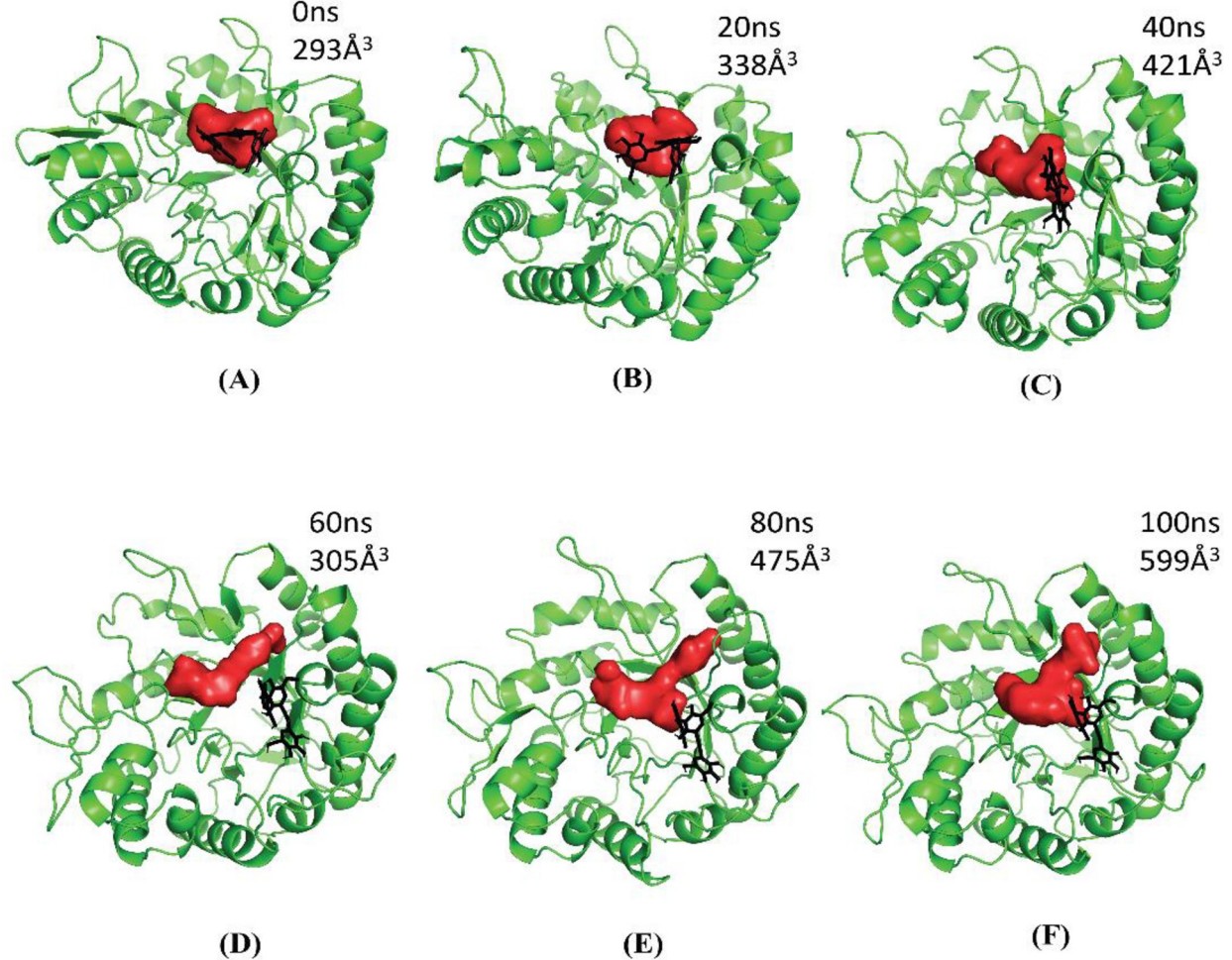

**Fig 6. The flexibility and volume analysis of substrate binding pocket of *βManAo* bound with M3.** Corresponding time intervals of (A) 0 ns, (B) 20 ns, (C) 40 ns, (D) 60 ns, (E) 80 ns and (F) 100 ns.

useful in comparing the differences in kinetic and thermodynamic behavior between two forms of proteins.

**3.6.8 Binding energy calculation.** Four different energy components for the ligand mannotriose towards the *βManAo* binding affinity were calculated. $\Delta G_{vdW}$ for the M3 was found to be ~ -20 kcal/mol; whereas the $\Delta G_{elec}$ value was -115 kcal/mol, which significantly contributed to the binding of the complex (Fig 10A). Comparison of the free energy components indicated that electrostatic energy is the main driving force behind the binding of mannotriose to *βManAo*. In general, the complex structure has a low binding free energy ($\Delta G_{bind}$ ~ -24 kcal/mol) and it is relatively stable. Enthalpy of binding ($\Delta H$) indicated the change in energy after binding of the ligand and was calculated as ~ -38 kcal/mol (Fig 10B). The high negative value of binding enthalpy reflected the formation of energetically more favorable non-covalent interactions between *βManAo* and mannotriose. The overall–$T\Delta S$ value was positive, indicating that the entropy of the system is negative, as the degree of freedom of the overall system had decreased. It was concluded that the complex was stable on the basis of heat energy distribution over the overall thermodynamic system. Per-residue energy decomposition in Fig 10C showed that Glu244 had high negative value of free energy contribution and actively participated for binding with M3 over the run. In addition, Ser286, Tyr283, Trp287 and Trp153 also

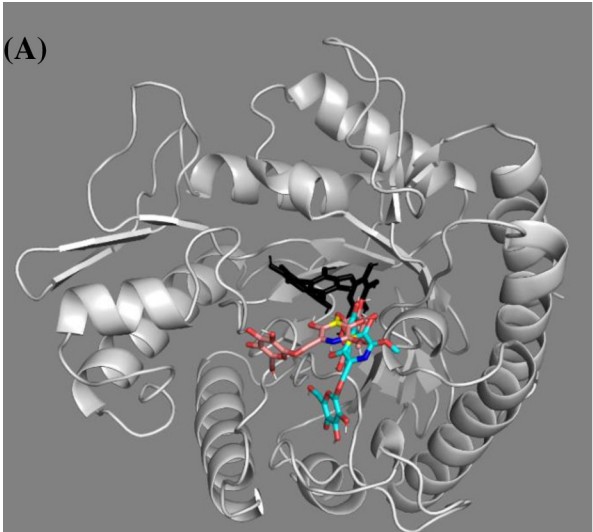
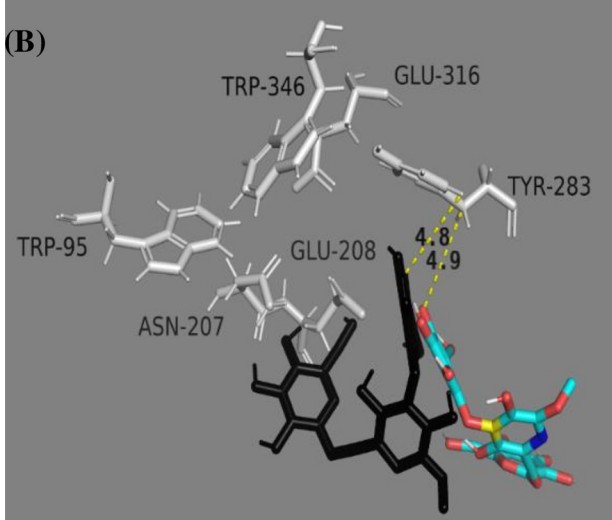
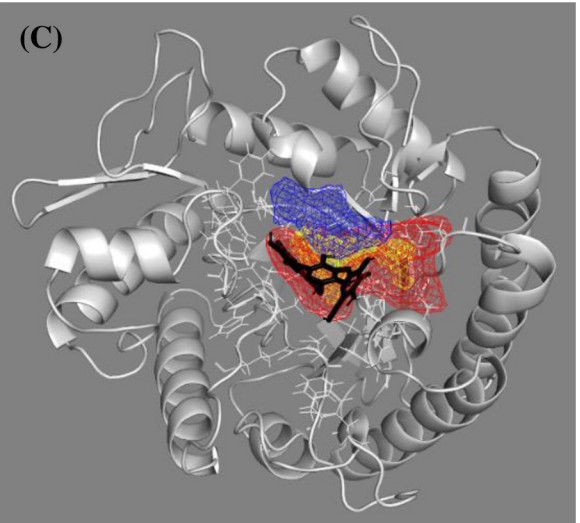

**Fig 7. The overall transition and interaction of M3 with *βManAo*.** (A) The transition of M3 from the initial docked position (black) to the final position (cyan) in the binding pocket of *βManAo*, (B) the interaction of M3 with the binding site residues (white) and anchored position of initial and final state of M3 with residue Tyr283 showing $\pi-\pi$ intercations at a distance of 4.8 and 4.9 Å, respectively. In (C), the overall pocket dynamics of *βManAo* in complex with M3 is shown. The average pocket over the simulation is marked in yellow, appearing pocket in red, disappearing pocket in blue and M3 in black.

favored binding. Fig 10D shows the per-residue per frame energy contribution of actively participating amino acids present in the catalytic cavity of *βManAo*. All the residues had favorable binding energy, except Glu208 and Arg210, which were binding unfavorably with the mannotriose (red sections in Fig 10D).

**3.6.9 Secondary structure analysis of proteins.** Define Secondary Structure of Proteins (DSSP) analysis elucidated the different secondary structure elements, such as alpha helices, beta sheets, and coils throughout the dynamic simulation. The results suggested that the binding of mannotriose to *βManAo*, in particular with Glu244, changed the secondary structure

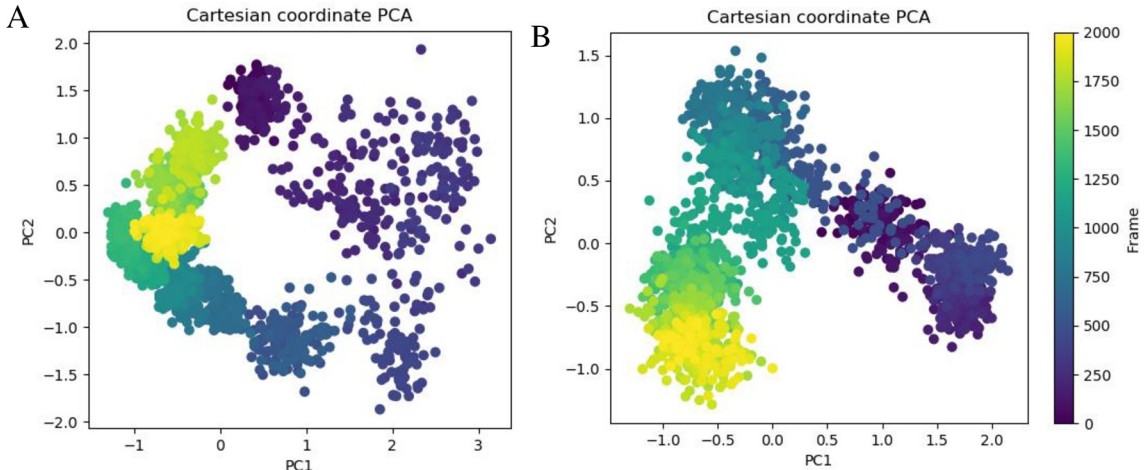

**Fig 8. Principal component analysis (PCA) plot of C$_\alpha$ atoms created by plotting first two eigenvectors in conformational space.**
(A) PCA plot of *βManAo* and (B) PCA plot of *ManAo*-M3 complex.

from an alpha helix to turn and vice versa transition between 20 and 60 ns (Fig 11). Interestingly, in case of transition for the Tyr283 containing secondary structure, the transition from alpha helix to turn was constant from about 50 to 100 ns (Fig 11).

The possible reason for this may be help turn the protein to pack the side chain and local environment for more stable conformation. It also yielded strong conformational propensities. The change to β-turn assists the protein to be more surface exposed for suitable interaction with mannotriose [49]. Turn plays a key role in the binding to the ligand and provides thermostability to the protein. For example, when a central turn in plastocyanin was mutated in a combinatorial fashion, it lost its binding ability towards its metal cofactor [50]. So, the helix-turn transition in the case of the *βManAo*-M3 complex favors thermodynamic stability and acts passively in folding the protein [49].

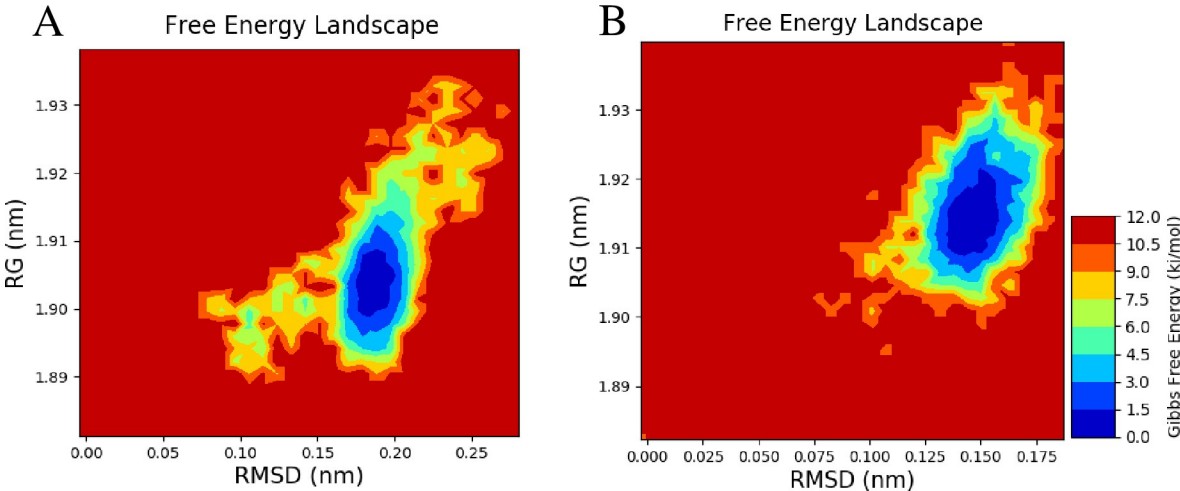

**Fig 9. Contour maps of free energy landscape as a function of RMSD (nm) value and radius of gyration (RG, nm).** (A) *βManAo* and (B) *βManAo*-M3 complex. The *βManAo*-M3 complex had lower subset of energy to reach the favorable state where the only enzyme *βManAo* had more than one energy subset.

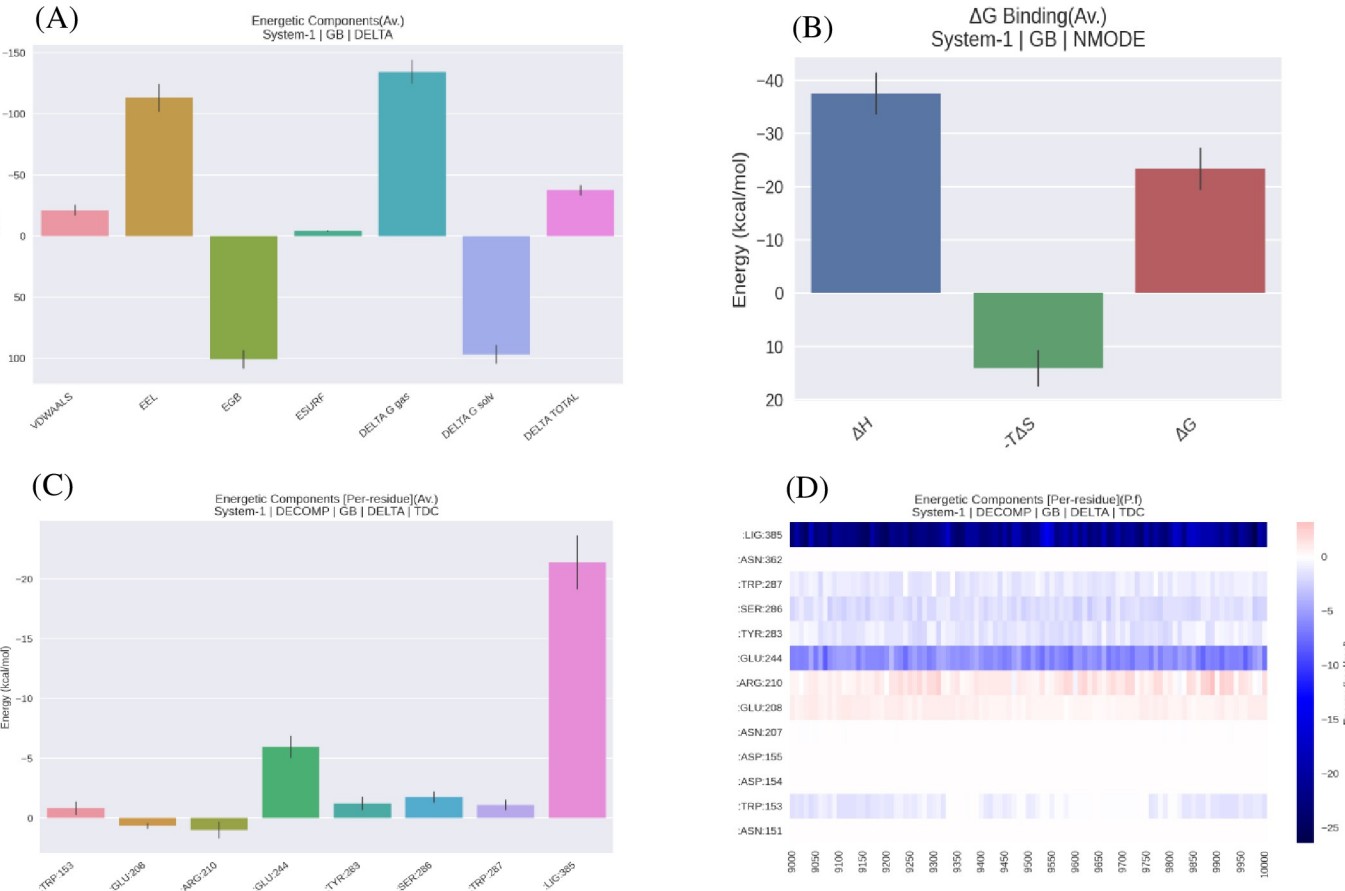

**Fig 10. Binding free energy analysis of *βManAo*-M3 complex.** (A) Enthalpic contributions to the *βManAo*-M3 interactions, (B) Total binding free energy ($\Delta G_{bind}$) for *βManAo*-M3 complex, (C) Per residue contribution in the binding free energy of M3 with *βManAo*, (D) Total decomposition of free energy contribution per residue and per frame of MD stimulation run (90-100ns). LIG385 refers to M3.

## 4. Conclusion

In the present study, the hypothetical model preparation and binding mechanism of *A. oryzae* β-mannanase with mannotriose was reported. All validated scores indicated that the modeled protein was correct for the molecular interaction. Docking analysis revealed that mainly polar amino acids like Glu208 and Glu244 play a key role at the binding site of the enzyme. However, other amino acids, namely Asn207, Asp155 and Ser286, are also important for the binding with the mannotriose. The docking results are validated through MD simulations and MMGBSA calculations, which reveal that the actual Glu244 residue is involved in the main catalytic reaction. The binding pocket of *βManAo* displayed a range of flexibility and change in volume to attune with the M3 displacement. From the volume of the substrate binding cavity estimated in this study, it is predicted that the enzyme could accommodate substrate molecules from M3 to M6. Metadynamic analysis showed the thermodynamic characters of protein and the protein ligand complex. Further, a detailed meta-dynamic analysis showed the change in secondary structure from helix to β-turn at the binding site for stabilizing the backbone hydrogen bonds for binding with ligand. Thus, the present study illustrated the structural properties of a prominent mannanase from a commonly explored fungus, which provides insights for improving the enzyme through protein engineering for the efficient degradation of mannans.

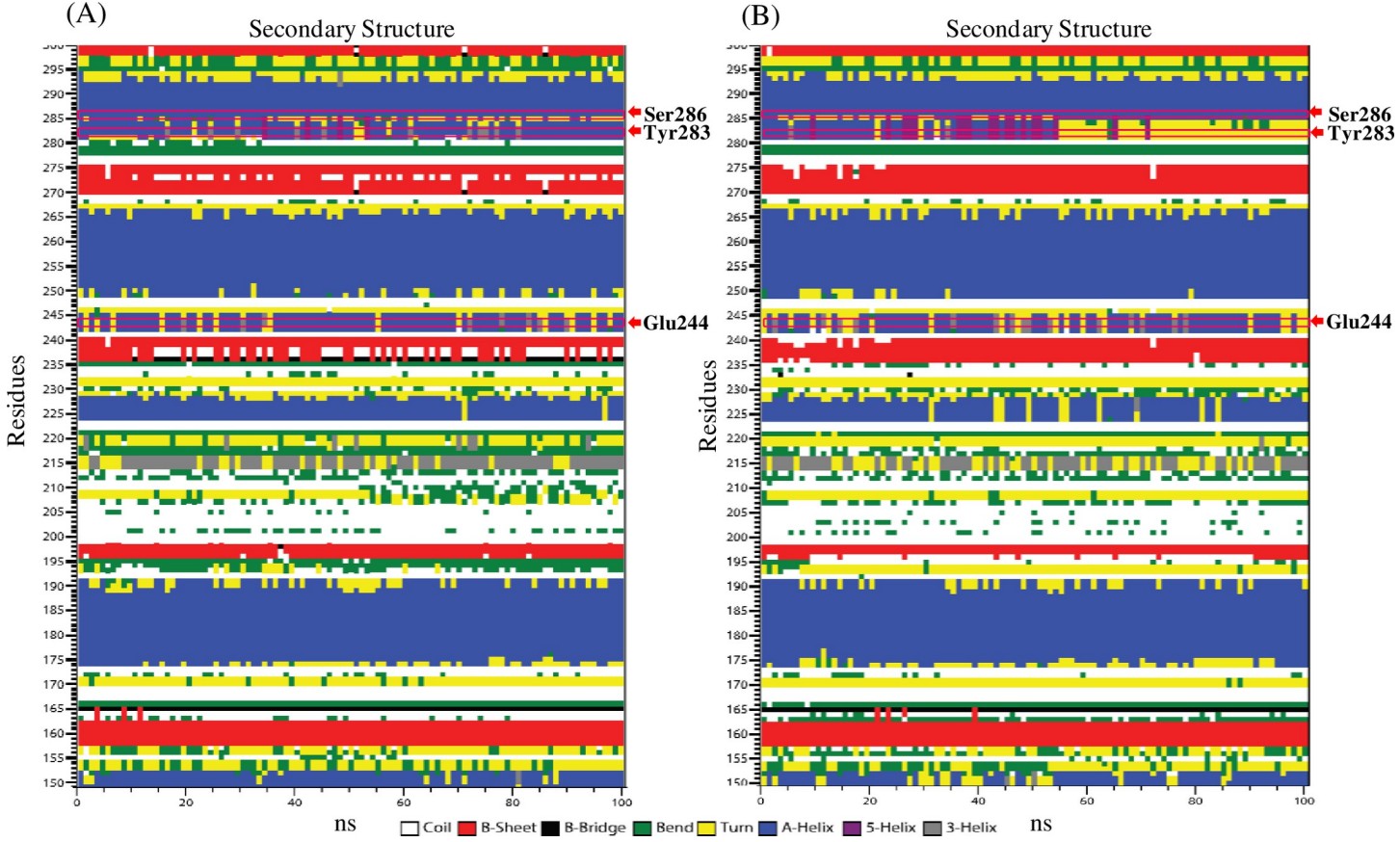

**Fig 11. Residue wise secondary structure of β-mannanase.** (A) *βManAo* and (B) *βManAo*-M3 during 100 ns dynamic simulation. The secondary structures were labeled as different color and three main binding amino acids over the simulation run (Glu244, Tyr283 and Ser286) were denoted by red arrow. A helix-turn transition was observed in Glu244 and Tyr283 amino acids.

## Supporting information

**S1 Table. Interaction of *βManAo* amino acids with substrate (M3) involved in different chemicals bonds.**
(DOCX)

## Acknowledgments

The infrastructural support of sophisticated instrumentation center and DST-PURSE (II) at Dr. Harisingh Gour Vishwavidyalaya, Sagar, MP, India is duly acknowledged. We would like to thank Rhodes University for APC. Author UKJ acknowledges ICMR, New Delhi for student fellowship.

## Author Contributions

**Conceptualization:** Uttam Kumar Jana, Naveen Kango.

**Data curation:** Uttam Kumar Jana, Gagandeep Singh.

**Formal analysis:** Uttam Kumar Jana, Gagandeep Singh.

**Investigation:** Uttam Kumar Jana, Gagandeep Singh.

**Methodology:** Uttam Kumar Jana, Gagandeep Singh.

**Project administration:** Uttam Kumar Jana, Naveen Kango.

**Resources:** Hemant Soni, Brett Pletschke, Naveen Kango.

**Software:** Uttam Kumar Jana.

**Supervision:** Brett Pletschke, Naveen Kango.

**Validation:** Uttam Kumar Jana, Hemant Soni, Naveen Kango.

**Visualization:** Uttam Kumar Jana, Gagandeep Singh.

**Writing – original draft:** Uttam Kumar Jana.

**Writing – review & editing:** Uttam Kumar Jana, Gagandeep Singh, Hemant Soni, Brett Pletschke, Naveen Kango.

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
