## [Decision Letter · Decision Letter 0]

17 Jan 2022

PONE-D-21-37808Molecular insight into Aspergillus oryzae β-mannanase interacting with mannotriose revealed by molecular dynamic simulation studyPLOS ONE

Dear Dr. Kango,

Thank you for submitting your manuscript to PLOS ONE. After careful consideration, we feel that it has merit but does not fully meet PLOS ONE’s publication criteria as it currently stands. Therefore, we invite you to submit a revised version of the manuscript that addresses the points raised during the review process.Please ensure that your decision is justified on PLOS ONE’s publication criteria and not, for example, on novelty or perceived impact.

We look forward to receiving your revised manuscript.

Kind regards,

Sabato D'Auria

Academic Editor

PLOS ONE

Journal Requirements:

(Author UKJ is grateful to ICMR, New Delhi for providing financial assistance as Senior Research Fellow. Authors thank Sophisticated Instrumentation Centre (SIC), Dr. Harisingh Gour Vishwavidyalaya, Sagar and DST PURSE (II) scheme for instrumentation facilities and financial support. Authors acknowledge the HPC facility of Indian Institute of Technology, Delhi for providing the computational resources for running the MD simulations.)

(The author(s) received no specific funding for this work.)

Reviewers' comments:

Reviewer's Responses to Questions

**Comments to the Author**

1. Is the manuscript technically sound, and do the data support the conclusions?

Reviewer #1: Yes

Reviewer #2: Yes

2. Has the statistical analysis been performed appropriately and rigorously? 

Reviewer #1: No

Reviewer #2: Yes

3. Have the authors made all data underlying the findings in their manuscript fully available?

Reviewer #1: Yes

Reviewer #2: Yes

4. Is the manuscript presented in an intelligible fashion and written in standard English?

Reviewer #1: Yes

Reviewer #2: No

5. Review Comments to the Author

Reviewer #1: The manuscript by Naveen Kango et al. entitled “Molecular insight into Aspergillus oryzae β-mannanase interacting with mannotriose revealed by molecular dynamic simulation study” reports an in-silico approach to study structural and functional aspects of an Aspergillus oryzae β- mannanase (βManAo). Fungal β-mannanases, hydrolyze β-1, 4-glycosidic bonds of mannans and find application in the generation of mannose and prebiotic mannooligosaccharides (MOS). The authors, through homology modeling and secondary structure prevision, created a βManAo model, that was validated and used for the docking, molecular dynamics, and secondary structure prevision studies. From the results obtained, were identified some features of this enzyme, like the binding site, the most important amino acid (Glu244 was found to play a key role in interaction with mannotriose, indicating conserved amino acid for the catalytic reaction), A detailed PCA analysis revealed the very flexible nature of the protein and energy landscapes suggested high conformation sub-states and complex dynamic behavior of the protein. DSSP analysis revealed the major transition of the protein from helix to turn when binding the mannotriose. The authors conclude that a hypothetical model was created, the binding mechanism of A. oryzae β-mannanase with mannotriose was identified where the Glu208 and Glu244 are the key amino acids at the active binding site, and also other amino acids (Asn207, Asp155, and Ser286) are important for the binding with the mannotriose.

The manuscript in the present form demands a revision before it can be published, so this reviewer suggests a “minor revision” of the paper.

Major issue:

1) The abstract is not very attractive; it should arouse curiosity in the reader and should be clearer. It should include details about the methodologies used to reach the results and in brief some details about the results in terms of numbers.

2) The results sub-section about the docking should be extended.

3) About the conclusion section: it is too simple, in my opinion, needs some improvements, as well as, add all the results obtained and a small perspective paragraph.

Minor issue:

1) Please try to improve, in general, the readability of the text helps the reader to better understand the paper.

2) All paper needs an accurate English revision and an accurate formatting check.

3) In Figure 1, panels A and E should be improved in terms of quality.

Reviewer #2: In this study, the authors show their results, using in silico approach of structural and functional aspects of an Aspergillus oryzae β- mannanase (βManAo). The manuscript is the interest to the journal, but I suggest publishing after minor revision” of the paper. In particular:

1. The abstract is not clear enough. I suggest the authors rewrite and improve.

2. In my opinion, are needed to improve the conclusion section of the manuscript. In particular, the authors should stress how their results can improve the know-how in the mannanase enzyme and their application in the degradation of mannanase.

3. The quality of Figure 1, panels A and E, is too low. Please, the authors should provide it.

4. The figure legends need more details. Please, the authors should provide it.

5. Accurate English revision of the text is needed.

6. PLOS authors have the option to publish the peer review history of their article (what does this mean?). If published, this will include your full peer review and any attached files.

Reviewer #1: No

Reviewer #2: No

---

## [Author Response · Author response to Decision Letter 0]

15 Mar 2022

Journal Requirements:

Reply: Revised as suggested

Reply: Revised as suggested

Reply: No, All relevant data are within the paper and its Supporting Information files.

(Author UKJ is grateful to ICMR, New Delhi for providing financial assistance as Senior Research Fellow. Authors thank Sophisticated Instrumentation Centre (SIC), Dr. Harisingh Gour Vishwavidyalaya, Sagar and DST PURSE (II) scheme for instrumentation facilities and financial support. Authors acknowledge the HPC facility of Indian Institute of Technology, Delhi for providing the computational resources for running the MD simulations.) We note that you have provided funding information that is not currently declared in your Funding Statement. However, funding information should not appear in the Acknowledgments section or other areas of your manuscript. We will only publish funding information present in the Funding Statement section of the online submission form. Please remove any funding-related text from the manuscript and let us know how you would like to update your Funding Statement. Currently, your Funding Statement reads as follows: (The author(s) received no specific funding for this work.)

Reply: 

The funding information regarding ‘financial assistance as Senior Research Fellow to UKJ’ has now been removed from the Acknowledgments section. 

 Reply: Not applicable

Reviewers' comments:

Reviewer's Responses to Questions

Comments to the Author

1. Is the manuscript technically sound, and do the data support the conclusions?

Reviewer #1: Yes

Reviewer #2: Yes

Reply: We are thankful to the reviewers for their kind comments. 

2. Has the statistical analysis been performed appropriately and rigorously?

Reviewer #1: No

Reviewer #2: Yes

Reply: We have again checked the requirement for statistical analysis, where relevant. 

3. Have the authors made all data underlying the findings in their manuscript fully available?

Reviewer #1: Yes

Reviewer #2: Yes

Reply: We are thankful to the reviewers for their kind comments. 

4. Is the manuscript presented in an intelligible fashion and written in standard English?

Reviewer #1: Yes

Reviewer #2: No

Reply: We are thankful to the reviewers for their kind comments. As per the comment of Reviewer #2, we have now revised the manuscript thoroughly with the help of a native English speaker. 

5. Review Comments to the Author

Reviewer #1: 

The manuscript by Naveen Kango et al. entitled “Molecular insight into Aspergillus oryzae β-mannanase interacting with mannotriose revealed by molecular dynamic simulation study” reports an in-silico approach to study structural and functional aspects of an Aspergillus oryzae β- mannanase (βManAo). Fungal β-mannanases, hydrolyze β-1, 4-glycosidic bonds of mannans and find application in the generation of mannose and prebiotic mannooligosaccharides (MOS). The authors, through homology modeling and secondary structure prevision, created a βManAo model, that was validated and used for the docking, molecular dynamics, and secondary structure prevision studies. From the results obtained, were identified some features of this enzyme, like the binding site, the most important amino acid (Glu244 was found to play a key role in interaction with mannotriose, indicating conserved amino acid for the catalytic reaction), A detailed PCA analysis revealed the very flexible nature of the protein and energy landscapes suggested high conformation sub-states and complex dynamic behavior of the protein. DSSP analysis revealed the major transition of the protein from helix to turn when binding the mannotriose. The authors conclude that a hypothetical model was created, the binding mechanism of A. oryzae β-mannanase with mannotriose was identified where the Glu208 and Glu244 are the key amino acids at the active binding site, and also other amino acids (Asn207, Asp155, and Ser286) are important for the binding with the mannotriose.

Reply: We are thankful to the reviewers for their kind comments. 

The manuscript in the present form demands a revision before it can be published, so this reviewer suggests a “minor revision” of the paper.

Major issue:

1) The abstract is not very attractive; it should arouse curiosity in the reader and should be clearer. It should include details about the methodologies used to reach the results and in brief some details about the results in terms of numbers.

Reply: We thank the reviewer for this very important comment. We have now revised the abstract in order to make it clearer and more interesting for the reader. We have also included details about the methodologies used and provided some brief details of results in numbers. 

2) The results sub-section about the docking should be extended.

Reply: As suggested, the results sub-section about the docking has now been extended. The extended section is now named as Section 2.5 and Section 2.6.

3) About the conclusion section: it is too simple, in my opinion, needs some improvements, as well as, add all the results obtained and a small perspective paragraph.

Reply: We have revised the conclusion and added some results, as well as a perspective paragraph. Line 426-427 and Line 432-434

Minor issue:

1) Please try to improve, in general, the readability of the text helps the reader to better understand the paper.

Reply: We have revised the text thoroughly so as to improve the readability of the paper. 

2) All paper needs an accurate English revision and an accurate formatting check.

Reply: We have revised the text in terms of the use of the English language and formatting as suggested. 

3) In Figure 1, panels A and E should be improved in terms of quality.

Reply: As suggested, the Figure 1, panels A and E are improved in terms of quality now. 

Reviewer #2: 

In this study, the authors show their results, using in silico approach of structural and functional aspects of an Aspergillus oryzae β- mannanase (βManAo). The manuscript is the interest to the journal, but I suggest publishing after minor revision” of the paper. In particular:

Reply: We thank the esteemed reviewer for going through the manuscript and kind comments. 

1. The abstract is not clear enough. I suggest the authors rewrite and improve.

Reply: We thank the reviewer for this comment. We have re-written the abstract to make it clearer. 

2. In my opinion, are needed to improve the conclusion section of the manuscript. In particular, the authors should stress how their results can improve the know-how in the mannanase enzyme and their application in the degradation of mannanase.

Reply: We have re-written the conclusion section to provide know-how on the mannanase enzymes and their application in the degradation of mannan.

3. The quality of Figure 1, panels A and E, is too low. Please, the authors should provide it.

 Reply: As suggested, the Figure 1, panels A and E are improved in terms of quality now. 

4. The figure legends need more details. Please, the authors should provide it.

Reply: Thank you for this very important comment. The figure legends have now been revised to render them complete and informative. 

5. Accurate English revision of the text is needed.

Reply: We have revised the English language in the text as suggested.

---

## [Editor Report · Decision Letter 1]

4 Apr 2022

PONE-D-21-37808R1Molecular insight into Aspergillus oryzae β-mannanase interacting with mannotriose revealed by molecular dynamic simulation studyPLOS ONE

Dear Dr. Naveen Kango

Thank you for submitting your manuscript to PLOS ONE. After careful consideration, we feel that it has merit but does not fully meet PLOS ONE’s publication criteria as it currently stands. Therefore, we invite you to submit a revised version of the manuscript that addresses the points raised during the review process. In particular, as required by PLOS policy, the authors should submit a full version of the original submission (included figures and legends to figures) as well as a full version of the revised manuscript (included figured and legend to figures).

We look forward to receiving your revised manuscript.

Kind regards,

Sabato D'Auria

Academic Editor

PLOS ONE

Journal Requirements:

Additional Editor Comments (if provided):

As required by PLOS policy, the authors should include the full version of the original submission (included Figures and legends to figures) as well as the full version of the revised manuscript (included figures and legends to figures).

---

## [Author Response · Author response to Decision Letter 1]

19 Apr 2022

1. Please amend your authorship list in your manuscript file to include author Brett Pletschke.

Reply:

Author Brett Pletschke is added now in the MS.

As per the instructions received from the Academic editor, original manuscript with figures and revised marked manuscript with figures is being submitted for kind consideration.

Reviewer #1: 

The manuscript by Naveen Kango et al. entitled “Molecular insight into Aspergillus oryzae β-mannanase interacting with mannotriose revealed by molecular dynamic simulation study” reports an in-silico approach to study structural and functional aspects of an Aspergillus oryzae β- mannanase (βManAo). Fungal β-mannanases, hydrolyze β-1, 4-glycosidic bonds of mannans and find application in the generation of mannose and prebiotic mannooligosaccharides (MOS). The authors, through homology modeling and secondary structure prevision, created a βManAo model, that was validated and used for the docking, molecular dynamics, and secondary structure prevision studies. From the results obtained, were identified some features of this enzyme, like the binding site, the most important amino acid (Glu244 was found to play a key role in interaction with mannotriose, indicating conserved amino acid for the catalytic reaction), A detailed PCA analysis revealed the very flexible nature of the protein and energy landscapes suggested high conformation sub-states and complex dynamic behavior of the protein. DSSP analysis revealed the major transition of the protein from helix to turn when binding the mannotriose. The authors conclude that a hypothetical model was created, the binding mechanism of A. oryzae β-mannanase with mannotriose was identified where the Glu208 and Glu244 are the key amino acids at the active binding site, and also other amino acids (Asn207, Asp155, and Ser286) are important for the binding with the mannotriose.

Reply: We are thankful to the reviewers for their kind comments. 

The manuscript in the present form demands a revision before it can be published, so this reviewer suggests a “minor revision” of the paper.

Major issue:

1) The abstract is not very attractive; it should arouse curiosity in the reader and should be clearer. It should include details about the methodologies used to reach the results and in brief some details about the results in terms of numbers.

Reply: We thank the reviewer for this very important comment. We have now revised the abstract in order to make it clearer and more interesting for the reader. We have also included details about the methodologies used and provided some brief details of results in numbers. 

2) The results sub-section about the docking should be extended.

Reply: As suggested, the results sub-section about the docking has now been extended. The extended section is now named as Section 2.5 and Section 2.6.

3) About the conclusion section: it is too simple, in my opinion, needs some improvements, as well as, add all the results obtained and a small perspective paragraph.

Reply: We have revised the conclusion and added some results, as well as a perspective paragraph. Section 4.

Minor issue:

1) Please try to improve, in general, the readability of the text helps the reader to better understand the paper.

Reply: We have revised the text thoroughly so as to improve the readability of the paper. 

2) All paper needs an accurate English revision and an accurate formatting check.

Reply: We have revised the text in terms of the use of the English language and formatting as suggested. 

3) In Figure 1, panels A and E should be improved in terms of quality.

Reply: As suggested, the Figure 1, panels A and E are improved in terms of quality now. 

Reviewer #2: 

In this study, the authors show their results, using in silico approach of structural and functional aspects of an Aspergillus oryzae β- mannanase (βManAo). The manuscript is the interest to the journal, but I suggest publishing after minor revision” of the paper. In particular:

Reply: We thank the esteemed reviewer for going through the manuscript and kind comments. 

1. The abstract is not clear enough. I suggest the authors rewrite and improve.

Reply: We thank the reviewer for this comment. We have re-written the abstract to make it clearer. 

2. In my opinion, are needed to improve the conclusion section of the manuscript. In particular, the authors should stress how their results can improve the know-how in the mannanase enzyme and their application in the degradation of mannanase.

Reply: We have re-written the conclusion section to provide know-how on the mannanase enzymes and their application in the degradation of mannan.

3. The quality of Figure 1, panels A and E, is too low. Please, the authors should provide it.

 Reply: As suggested, the Figure 1, panels A and E are improved in terms of quality now. 

4. The figure legends need more details. Please, the authors should provide it.

Reply: Thank you for this very important comment. The figure legends have now been revised to render them complete and informative. 

5. Accurate English revision of the text is needed.

Reply: We have revised the English language in the text as suggested.

---

## [Editor Report · Decision Letter 2]

28 Apr 2022

Molecular insight into Aspergillus oryzae β-mannanase interacting with mannotriose revealed by molecular dynamic simulation study

PONE-D-21-37808R2

Dear Dr.Naveen Kango,

We’re pleased to inform you that your manuscript has been judged scientifically suitable for publication and will be formally accepted for publication once it meets all outstanding technical requirements.

Kind regards,

Sabato D'Auria

Academic Editor

PLOS ONE
---

## [Editor Report · Acceptance letter]

19 Aug 2022

PONE-D-21-37808R2 

Molecular insight into *Aspergillus oryzae* β-mannanase interacting with mannotriose revealed by molecular dynamic simulation study 

Dear Dr. Kango:

I'm pleased to inform you that your manuscript has been deemed suitable for publication in PLOS ONE. Congratulations! Your manuscript is now with our production department. 

Kind regards, 

on behalf of

Dr. Sabato D'Auria 

Academic Editor

PLOS ONE